# Credibility assessment of *in silico* clinical trials for medical devices

Pras Pathmanathan[1]*, Kenneth Aycock[1], Andreu Badal[1], Ramin Bighamian[1], Jeff Bodner[2], Brent A. Craven[1], Steven Niederer[3,4]

1 Center for Devices and Radiological Health, US Food and Drug Administration, Silver Spring, Maryland, United States of America, 2 Medtronic, PLC., Minneapolis, Minnesota, United States of America, 3 National Heart and Lung Institute, Imperial College, London, United Kingdom, 4 The Alan Turing Institute, London, United Kingdom

* pras.pathmanathan@fda.hhs.gov

**Data Availability Statement:** All relevant data are within the paper and its Supporting information files.

## Abstract

*In silico* clinical trials (ISCTs) are an emerging method in modeling and simulation where medical interventions are evaluated using computational models of patients. ISCTs have the potential to provide cost-effective, time-efficient, and ethically favorable alternatives for evaluating the safety and effectiveness of medical devices. However, ensuring the credibility of ISCT results is a significant challenge. This paper aims to identify unique considerations for assessing the credibility of ISCTs and proposes an ISCT credibility assessment workflow based on recently published model assessment frameworks. First, we review various ISCTs described in the literature, carefully selected to showcase the range of methodological options available. These studies cover a wide variety of devices, reasons for conducting ISCTs, patient model generation approaches including subject-specific versus 'synthetic' virtual patients, complexity levels of devices and patient models, incorporation of clinician or clinical outcome models, and methods for integrating ISCT results with real-world clinical trials. We next discuss how verification, validation, and uncertainty quantification apply to ISCTs, considering the range of ISCT approaches identified. Based on our analysis, we then present a hierarchical workflow for assessing ISCT credibility, using a general credibility assessment framework recently published by the FDA's Center for Devices and Radiological Health. Overall, this work aims to promote standardization in ISCTs and contribute to the wider adoption and acceptance of ISCTs as a reliable tool for evaluating medical devices.

## Author summary

A new method for evaluating a medical device is an *in silico* clinical trial, where the device performance is tested using computational models representing a cohort of patients. Demonstrating that *in silico* clinical trials are reliable is clearly of paramount importance. However, little information is available on how to ensure this. In this paper we present a workflow for evaluating an *in silico* clinical trial. This workflow was developed by considering a wide-ranging sample of previously published *in silico* clinical trials, together with a

**Funding:** The author(s) received no specific funding for this work.

**Competing interests:** The authors have declared that no competing interests exist.

detailed assessment of how specific model evaluation activities apply to *in silico* clinical trials.

## 1. Introduction

*In silico* clinical trials (ISCTs) are an emerging method in computational modeling and simulation (M&S) where medical interventions are evaluated using a range of computational models of patients. ISCTs have many potential applications including reducing animal testing, augmenting or reducing the size of real-world clinical trials (CTs), supporting trial design by providing improved inclusion-exclusion criteria, or effectively increasing the diversity of patients considered in the overall safety and effectiveness assessment compared to a traditional CT. Realizing the potential of ISCTs will require addressing technical challenges in model development, data collection, and computational methods. This paper focuses on another challenge hindering the use of ISCTs: establishing processes and standard approaches to demonstrate the credibility of their results.

Model credibility is defined as the "trust in the predictive capability of a computational model for a particular context of use (COU)" [1]. The medical devices modeling community has developed frameworks and methods to assess credibility of computational models for medical devices, including the ASME V&V40 2018 Standard, FDA Guidance [2], and various supporting documents [3–7]. These efforts build upon extensive verification, validation, and uncertainty quantification (VVUQ) literature developed by the engineering community [8]. In the context of M&S, verification confirms that a computational model accurately represents an underlying mathematical model, while validation involves comparing model predictions with real-world data. Uncertainty quantification (UQ) entails estimating uncertainty in model inputs and computing the subsequent uncertainty in model outputs [2].

Most recent medical device credibility efforts have been primarily motivated by traditional applications of M&S for medical devices, such as identifying worst-case conditions to reduce the burden of bench testing. ISCTs and a closely related counterpart, patient-specific models (PSMs), have received limited investigation in comparison. PSMs are computational patient models in which some parameters have been personalized using data from a specific patient, so that the model can make predictions for that patient. In previous work, we explored credibility of patient-specific models [9]. We considered what verification, validation and uncertainty quantification each mean for PSMs, unique considerations that arise for PSM credibility assessment, and discussed how to apply ASME V&V40 2018 to PSMs. The present paper extends the previous work to *in silico* clinical trials. The aims of the present paper are: (i) consider what verification, validation and uncertainty quantification mean for ISCTs; (ii) identify unique considerations that arise when evaluating credibility of an ISCT; and (iii) propose a credibility assessment workflow for ISCTs based on FDA Guidance [2] and ASME V&V40 2018 [1]. In this paper our focus is ISCTs for medical devices; however, our conclusions may inform evaluation of ISCTs in other domains, such as ISCTs for pharmaceutical products. This paper complements and builds upon related work, including a theoretical framing for credibility of *in silico* studies [10], a discussion of limitations of ASME V&V40 2018 for ISCTs [11], a framework for executing an ISCT using VVUQ concepts [12], and a proposed alteration to ASME V&V40 2018 for ISCTs, including modification of the risk assessment stage which is outside the scope of the present paper [13]. The present paper focuses primarily on first-principles models, as opposed to data-driven models (which includes those based on

machine learning). This will be discussed further in the following section. For a review of ISCTs for medical imaging devices focused primarily on data-driven models, see [14].

Our approach is as follows. First, we review a selection of ISCTs described in the literature that cover a wide range of possibilities. The goal of this literature review is not to comprehensively survey all ISCTs that have been performed for evaluating medical devices. Instead, it is to illuminate the array of methodological options with ISCTs. This will inform the credibility assessment discussion in subsequent sections. We begin in Section 2 by discussing what an ISCT is, defining sub-models of ISCTs, and reviewing example ISCTs. From these examples, we identify unique VVUQ considerations for ISCTs in Section 3 and propose an ISCT credibility assessment workflow in Section 4.

## 2. *In silico* clinical trials for medical devices

### 2.1. Definitions

There is currently no widely accepted community definition of an ISCT, and interpretations of ISCTs differ among practitioners. In fact, related terms such as *in silico* trial [10,15,16] or *in silico* study (thousands of publications) are used in the M&S literature, and there is no clear delineation between the three terms '*in silico* trial', '*in silico* study' and '*in silico* clinical trial'. For the purposes of this document, we will interpret an ISCT as follows: a computational study in which a medical intervention is evaluated using a cohort of computational models of patients. In this paper, the medical intervention will be a medical device, including imaging and other diagnostic tools as well as therapeutic devices.

Requiring a medical intervention in the definition distinguishes ISCTs from simulation studies addressing questions of basic physiology or pathophysiology only, with no medical intervention. Such simulation studies are analogous to observational clinical studies. Our interpretation of ISCTs imposes no constraints on the complexity of either the device model or patient model. Studies using very simple device or patient representations fall within our interpretation of an ISCT. We also do not impose constraints on the number or variability of the patients simulated. While ideally these would cover the indicated patient population in some way, with sufficiently many virtual patients to draw statistically significant conclusions, we do not require these in our interpretation of ISCTs (similarly to the definition of a clinical trial [17]). Note also that we have not constrained the type of question to be addressed in the ISCT. It is possible that a future consensus definition of ISCT will require the question of interest to be a specific safety or effectiveness question (e.g., related to a clinical endpoint), or perhaps a regulatory question. However, a broad interpretation of ISCTs is sufficient for this paper.

We will use the following terminology. A **patient model** is a computational model of a patient for a biomedical application. A **baseline patient model** is a patient model in which values of some parameters (referred to as **fixed parameters**) have been specified but others (referred to as **variable parameters** or **personalized parameters**) have not yet been specified. The baseline patient model has also been referred to as the template model [18]. A **virtual patient** is a realization of the baseline patient model, that is, a model in which the variable parameters have been specified. A **subject-specific virtual patient** is a virtual patient for which the values of the variable parameters have been specified based on data from a real (living or deceased) human subject. Subject-specific virtual patients can be considered to correspond to that real subject. A **synthetic patient** is a virtual patient where the values of the variable parameters do not, or are not known to, correspond to any real human subject–typically parameters obtained by sampling some distribution. A **virtual cohort** is a set of virtual patients. Virtual cohorts may be made entirely of subject-specific virtual patients, entirely of synthetic patients, or could perhaps be a mixture of both. An ISCT is a simulation study using

a virtual cohort addressing a specific question of interest about a medical intervention. For the remainder of this paper, when we refer to an ISCT, we mean an ISCT for a medical device.

In healthcare applications, the term 'digital twin' is sometimes used synonymously with patient-specific model. Alternatively, it has been defined more precisely as a "comprehensive, virtual tool that integrates coherently and dynamically the clinical data acquired over time for an individual using mechanistic and statistical models" [19], that is, a patient-specific model that is updated as new data from the patient becomes available. *In silico* clinical trials using virtual cohorts of digital twins are possible and are a special case of virtual cohorts of subject-specific virtual patients.

It will also be important to distinguish between two different modeling modalities. **First principles-based models** (sometimes referred to as knowledge-based models [20]) are models constructed around cause-effect hypotheses encapsulated as physics-based, mechanistic or first principles based governing equations. These models explicitly integrate known physics and physiology. **Data-driven models** are not based on first principles governing equations. These models are often associated with fitting to large datasets. Modern machine learning methods are data-driven models, whereas traditional solid mechanics or fluid dynamics models are first principles-based (although they may have data-driven sub-models). This is not a binary categorization. Rather, many models lie within a spectrum with pure data-driven and pure first principles based models at the extremes [20], and hybrid methods are possible, e.g., [21]. Still, it is important to distinguish between the two approaches as different evaluation techniques apply.

## 2.2. Features of an ISCT

Fig 1 presents an overview of components to an ISCT. ISCTs are always composed of multiple sub-models. Based on our definitions above, the following are necessary sub-models for a medical device ISCT. Examples will be provided in Section 2.3.

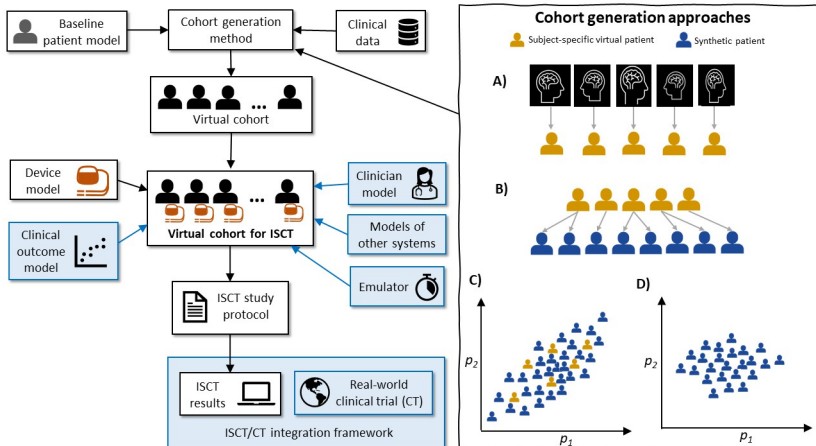

**Fig 1. Overview of components of an ISCT (necessary components in white boxes, possible components in light blue boxes).** Left side: a virtual cohort (set of virtual patients) can be created given a baseline patient model and a cohort generation method. Coupling with a device model, and potentially other sub-models, results in the virtual cohort for the ISCT, which is run using the ISCT study protocol to obtain the ISCT results. The right-hand box shows four approaches to generating a virtual cohort (not comprehensive). Approach A): Construct subject-specific virtual patients from subject-specific data. B) Construct subject-specific virtual patients and alter to create synthetic patients. C) Generate synthetic patients by sampling from probability distributions derived from subject-specific virtual patients; D) Generate synthetic patients by sampling from prescribed probability distributions. Two parameters, $p_1$ and $p_2$, shown for simplicity. See text for discussion.

1. **Baseline patient model**: as defined above. By patient model, we do not mean just an anatomical representation; also included are the governing equations for physical/physiological/clinical quantities of interest. The governing equations could be primarily first principles-based or primarily data-driven.

2. **Device model**: a model of the medical device under consideration, or some aspect of the device. As with the baseline patient model, by device model we refer to geometrical information about the device together with equations governing the device quantities of interest. Again, the governing equations could be primarily first principles-based or primarily data-driven. We assume the device will be simulated *in silico*. The case of hardware-in-the-loop testing, where a real device interacts with a patient model [15], is outside the scope of this paper.

3. **Virtual cohort**: the set of virtual patients to be used in the ISCT. Fig 1 distinguishes between two states, the virtual cohort before and after coupling with the device and other models. Many different methods could be used to specify variable parameters in the baseline patient model to create virtual patients that make up the virtual cohort. We highlight four approaches as illustrated in Fig 1:

   1. Construct $N$ subject-specific virtual patients using clinical data collected from $N$ subjects

   2. Construct $N$ subject-specific virtual patients using clinical data collected from $N$ subjects, then alter these models in some way (e.g., introduce disease conditions) to create $M \geq N$ synthetic patients (example in Section 2.3.5).

   3. Construct $N$ subject-specific virtual patients using clinical data collected from $N$ subjects and use the parameter values inferred across the $N$ subjects to set up probability distributions for the variable parameters. Next, sample from these distributions to generate $M$ synthetic patients (usually $M >> N$). The distributions could be multi-variate to account for parameter correlation, as indicated in Fig 1.

   4. Set up probability distributions using a different method to (c), for example assuming uniform or normal distributions for parameters with bounds/means/variances taken from the clinical literature, and then sample from those distributions to generate $M$ synthetic patients. Since there is generally limited information on parameter correlations across patients in the clinical literature, these distributions will likely be independent.

   Other approaches to generating virtual cohorts are possible, including variations of the above, or others such as [22] which constructs $N$ subject-specific virtual patients and then uses a combinatorial approach to create synthetic patients. To create a virtual cohort, it may be necessary to apply inclusion-exclusion criteria to a larger group of virtual patients. These criteria could be based on factors such as the anatomical [16] or functional features of the patient models.

   ISCTs may also contain the following sub-models, among others:

4. **Clinical outcome mapping model**: in many cases, virtual patient models predict an engineering quantity (e.g., stresses, strains) that differ from the clinical outcomes that are typically used as study endpoints, such as mortality, hospitalization, or sustained freedom from disease. ISCTs may therefore also use a clinical outcome mapping model (henceforth referred to a 'clinical outcome model') that converts the patient model output(s) into the clinical outcome(s). Such models are typically data-driven, based on empirical associations between the quantities. For example, [12] describes an ISCT which uses a statistical model

to relate the engineering output (displacement of a neuromodulation lead tip) with the clinical outcome (complication-free rate). Clinical outcomes are often binary or categorical, whereas model outputs are usually continuous variables, in which case the clinical outcome model may require mapping continuous variables to categorical ones.

5. **Model(s) of other relevant systems**: the system of interest may involve other sub-systems, excluding the patient and device, that need to be modeled in the ISCT. For example, an ISCT for assessing heating of a metallic implant during MR imaging requires a model of the MR scanner (see Section 2.3.1).

6. **Clinician model**: some ISCTs explicitly simulate the behavior of a clinician, for example a surgeon placing a device in a patient or a radiologist reviewing X-ray images (see Section 2.3.2). Additionally, whenever an ISCT involves an implanted device, simulations must account for where and how the device is implanted (e.g., following instructions for use), and clinician variability in the placement could be important to consider. As with hardware-in-the-loop studies, ISCTs with a "clinician-in-the-loop" (e.g., a real radiologist reviewing each simulated image) are outside the scope of this document.

7. **Emulator**: an emulator is a fast-running surrogate model used in lieu of a more computationally expensive model. ISCTs can use emulators that are trained on many simulations from virtual patients to provide fast predictions. Emulators are often statistical models, for example, Gaussian process emulators [23, 24], although they could also utilize lower-fidelity physics-based models [25].

Other components of ISCTs (not sub-models) illustrated in Fig 1 include the ISCT study protocol and any ISCT/CT integration framework. The ISCT/CT integration framework refers to any statistical framework for integrating an ISCT with a real-world clinical trial. For example, [26] proposes a workflow for integrating an ISCT and a CT, where the weight assigned to the ISCT is dictated by the agreement between the ISCT and CT—this is discussed further in Section 2.3.6. ISCT study protocol and ISCT/CT integration framework are both related to ISCT **execution**. The focus of this paper is ISCT credibility, and a detailed analysis of the options available for ISCT execution is outside our scope. However, we note there are many options available for ISCT study execution, including some that might not be feasible in a real-world study. For example, each virtual patient could receive both the control and intervention in different simulations. For the case of multiple interacting interventions, it may be possible to combinatorically consider all possibilities for each virtual patient. *Ad hoc* removal of virtual patients, for example based on non-physiological behavior, is also a possibility in ISCTs. The range of possibilities for ISCT execution is, to our knowledge, largely unexplored. This hinders the ability to develop of standardized processes for setting up and running an ISCT. Establishing good practice for all stages in an ISCT, of which the credibility assessment as covered below will be a subset, will be vital to the success of the field.

## 2.3. Example ISCTs for medical devices

Next, we review ISCTs that have been performed for medical device applications. Our goal is to demonstrate the range of methodological options available with ISCTs, not to provide a comprehensive review of all ISCTs performed for medical devices. Therefore, we have selected eight ISCTs that we believe showcase different applications, features, or decisions with ISCTs, that will inform the credibility assessment sections that follow. The eight examples are summarized in Table 1.

**Table 1. Overview and comparison of ISCTs discussed in Section 2.3.** MR: magnetic resonance, RF: radiofrequency, AUC: area under receiver operator characteristic curve, ODE: ordinary differential equation, CGM: continuous glucose monitor, FEA: finite element analysis.

| ISCT | Device evaluated | ISCT aim | Patient model | Device model | Virtual cohort generation approach | N | Other sub-models / components |
|---|---|---|---|---|---|---|---|
| MR induced heating (ISO 10974, clause 8, Tier 4 [28]) | Active implantable metallic device (AIMD) | Assess RF-induced heating of AIMD during MR imaging. Computed: local temperature rise | Electromagnetic whole-body model (3D, ~300 tissue regions, EM properties assigned to each region). | Electromagnetic model of the AIMD | Range of subject-specific models constructed from data from twelve patients, covering a range of ages, both sexes, body masses, and including pregnant women models. | 15 | (1) Model of RF transmit coil as source of induced electric field. (2) Convert RF power deposition into local temperature rise using, e.g., bioheat equation |
| VICTRE [32] | X-ray imaging devices for digital breast tomosynthesis (DBT) | Compare DBT and digital mammography (DM) images for lesion detection. Computed: AUC difference between DBT and DM | Breast mechanics model with randomly created anatomical structures, including microcalcification / speculated mass. Breast bio-mechanics model to simulate compression as done in clinical practice. | Physics-based model of images generated by DBT and DM systems | Synthetic patients with range of breast sizes and radiographic densities representative of the screening population. Multiple lesions inserted inside each patient. 2D and 3D regions of interest (with and without lesions) extracted for detection performance analysis. | 2,986 | Computational reader model (i.e., clinician model) |
| FD-PASS [16] | Intra-cranial flow diverter | Assess flow diverter efficacy for occluding aneurysms. Computed: aneurysm mean velocity reduction | Computational fluid dynamics model of carotid artery | Structural model of flow diverter | Initially >300 anatomical models generated from images, then inclusion-exclusion criteria applied (82 models), then normotensive and hypertensive conditions applied to each. | 164 | No clinical outcome model as such but aneurysm mean velocity reduction > 35% was considered a surrogate for successful occlusion. |
| Artificial pancreas ISCT [39] (example of PCLC evaluation) | Automated insulin delivery device | Assess stability of controller. Computed: glucose time profiles | Compartmental glucose-insulin model (13 ODEs, 26 free parameters (2008 version)) | CGM sensor and glucose pump models, and implemented controller algorithm | Parameters sampled from multi-variate distribution | 300 | |
| Whole heart model ISCT [41–43] | Cardiac resynchronization therapy (CRT) systems | Aim to compare the efficacy of different methods for delivering CRT. Computed quantity: Activation timings and synchrony | 4 chamber heart electrophysiology model | Different devices mimicked by changes in model boundary conditions. | Subject-specific models generated using cardiac images from 24 patients, then each model altered to introduce heart failure | 24 | |

*(Continued)*

**Table 1.** (Continued)

| ISCT | Device evaluated | ISCT aim | Patient model | Device model | Virtual cohort generation approach | N | Other sub-models / components |
|---|---|---|---|---|---|---|---|
| Bayesian framework for integrating ISCT and CT [26] | Implantable cardioverter defibrillator (ICD) | Assess lead fracture rate | Statistical model of bending stress across the population derived from imaging leads in real patients. (No single patient model) | Function relating bending stress to number of cycles to fracture | See patient model. | Not fixed | Bayesian statistical framework for incorporating real-world trial |
| Automata-based heart models for ICD evaluation [46] | Implantable cardioverter defibrillator (ICD) | Assess ability to discriminate two types of arrhythmias Computed: rate of inappropriate detection | Automata-based model of cardiac electrophysiology capable of producing 19 different arrhythmias | Two arrhythmia detection algorithms | Uniformly distributed timing parameters with ranges determined from EP literature | 11,400 | [47] extends this work by placing model within a Bayesian statistical framework including incorporating with a real-world trial |
| Micra | Micra implantable leadless pacemaker | Predict reliability of fixation tines for fatigue-related failures | Impact of the patient is accounted for via stochastic boundary-conditions. | Structural FEA model of the fixation tines | Stochastic generation of multiple virtual trials each with multiple virtual patients. | 10K simulated trials of 5K virtual patients each, 50M simulations total | Statistical model of fracture observed in human patients in real world trial via imaging to confirm rate of failure |

**2.3.1. MR-induced heating of implantable metallic devices.** Every year millions of patients receive life-long metallic implants such as orthopedic devices or pacemakers. Many of these patients will require MR imaging at some later point in their life. However, metallic implants may absorb radiofrequency (RF) energy emitted by the MR scanner which can cause thermal tissue damage. It is especially challenging to demonstrate that absorbed energy falls within safe limits for active metallic implants (e.g., cardiac pacemakers), which have leads that may take various possible configurations within the body. A well-established method addressing this safety question is computational electromagnetic modeling [27,28]. In this approach, an electromagnetic model is created of relevant aspects of the implant. The implant model is embedded in a patient model, coupled to a model of the MR scanner, and Maxwell's equations are solved to predict specific absorption rate (SAR; rate of RF energy deposited in tissue). An additional model, for example, the Pennes bioheat equation [29], can be used to convert SAR into a rise in temperature. The Virtual Population [30] is a set of human models for electromagnetic simulations that have been created for these purposes. Each model is made up of over 300 tissues, with a supporting database of electromagnetic properties for each tissue. Fifteen human models covering a range of ages, body mass indices, and sexes have been generated using data from 12 subjects (one of the subjects provides multiple models at different stages of pregnancy). While simulation studies involving the Virtual Family are generally not described as ISCTs, they fall within our interpretation of ISCTs in Section 2.1. For this application, every virtual patient is a subject-specific virtual patient; there are no synthetic patients.

**2.3.2. Virtual Imaging Clinical Trial for Regulatory Evaluation (VICTRE).** Nearly 40 million x-ray-based breast images are acquired every year in the United States. Two alternative imaging technologies are available in the market: 2D examinations with full-field digital mammography (FFDM) and 3D examinations with digital breast tomosynthesis (DBT). The clinical evaluation and comparison of x-ray imaging modalities is challenging due to the inherent risk of ionizing radiation, particularly for radiosensitive organs like the breast. Even though DBT

has been approved by the FDA since 2011, it is still considered a higher risk device (class III) than FFDM (class II), and a large clinical trial with more than 100,000 patients is underway to compare the two technologies (Tomosynthesis Mammographic Imaging Screening Trial [31]). To address the difficulty of evaluating these technologies, the Virtual Imaging Clinical Trial for Regulatory Evaluation (VICTRE) project [32] explored the possibility of using an ISCT to compare the performance of FFDM and DBT. First, a cohort of 3,000 synthetic breast phantoms was created by sampling a mathematical anatomic model. Then, FFDM and DBT images were simulated using a physics-based Monte Carlo x-ray transport code. Finally, the detectability of lesions inside the breasts was evaluated using mathematical model observers (i.e., a clinician model). The results of the ISCT compared favorably with the results of an existing clinical trial that was used as a reference, indicating that *in silico* imaging trials represent a viable source of regulatory evidence for imaging devices evaluation.

**2.3.3. Intracranial flow diverter performance assessment.** Flow diverters are stent-like devices implanted adjacent to intra-cranial aneurysms to promote thrombus formation in the aneurysm. The flow diverter performance assessment (FD-PASS) study [16] was an *in silico* clinical trial that predicted post-treatment flow reduction in 164 virtual patients using computational fluid dynamics (CFD). First, a reference cohort of over 300 anatomical models was generated from 3D rotational angiography images. 82 anatomies were retained after applying inclusion-exclusion criteria relating to aneurysm structure and poor mesh quality. Normotensive and hypertensive conditions were simulated for each of the 82 anatomies, generating 164 virtual patients in total. A virtual model of the Pipeline Embolization Device (Medtronic) was deployed in each virtual patient, following clinical guidelines. CFD simulations were performed to compute post-treatment aneurysm mean velocity reduction (AMVR). The ISCT endpoint was AMVR > 35%, argued as a surrogate for successful occlusion. Flow diverter efficacy as predicted in the FD-PASS trial was compared with observations from three previous clinical trials.

**2.3.4. Physiologic closed loop control (PCLC) device evaluation.** Physiologic closed loop control (PCLC) devices automatically actuate therapy based on data sensed from the patient to achieve a target response set by the user. Examples of PCLC devices include autonomous mechanical ventilators that constantly adjust a patient's end-tidal oxygen [33], automated anesthetic devices [34], automated fluid resuscitation devices [35], or automated insulin pumps that measure blood glucose levels and deliver insulin accordingly. PCLC algorithms need to demonstrate basic properties such as stability and robustness. Computational patient models of the physiological system(s) of interest can be used to evaluate a PCLC system. An example workflow is to develop a patient model, sample parameters across a wide range to generate a wide range of synthetic patients, and assess PCLC performance using each patient model [36]. Patient models used in PCLC evaluation are typically systems of ordinary differential equations (ODEs), such as reduced order ODEs governing whole-body hemodynamics [37]. Another example is compartmental glucose-insulin models, which have been used to generate virtual cohorts to evaluate automated insulin pumps [38,39] (more details in Table 1). PCLC evaluation provides an example of case C in Fig 1. For example, [37] defines a method for generating synthetic patients from calibrated subject-specific virtual patients, for a hemorrhage/fluid resuscitation model, that attempts to minimize sampling from non-physiologic parameter space.

**2.3.5. Evaluating cardiac devices using virtual cohorts of electrophysiological whole-heart models.** Patient-specific models of heart electrical activity can be constructed from MR or computerized tomography imaging data. Libraries of heart models have been developed that can be used as virtual cohorts for device or drug ISCTs. For example, imaging data from 24 subjects was used in [40] to develop 24 four-chamber heart models. These models were

used in [41–43] to study the efficacy of different cardiac resynchronization therapy (CRT) pacing protocols for heart failure patients with left bundle branch block (LBBB). The CRT device was not explicitly simulated; instead, CRT lead placements were simulated by prescribing stimuli locations in the electrophysiological simulations. In these studies, both subject-specific and synthetic cohorts were created. Each of the 24 anatomical models represent a specific patient, although tissue material properties were defined from the literature. Synthetic patients were then created from the subject-specific virtual patients, by introducing different presentations of electrophysiology pathologies in each patient (case B in Fig 1). The cohort of synthetic patients was used to study how different CRT delivery systems are affected by underlying pathology and highlighted the need for personalized therapy [44]. To validate the cohort, the authors compared cohort predictions for different pathology presentations and responses to therapy to summary statistics from independent clinical studies [45]. This provides a framework for validating predictions from synthetic virtual cohorts.

**2.3.6. Bayesian framework for incorporating virtual patient predictions in medical device clinical trials.**   The question of how to augment a real-world clinical trial with results from an ISCT is addressed in [26]. This paper assumes a computational model has been developed that predicts the same endpoint as a clinical trial, and that the CT and ISCT can be run in parallel. The specific application considered in [26] is cardiac lead fracture. However, the main contribution of [26] is a novel statistical framework, and we focus on the framework in this summary. The article presents a Bayesian statistical framework for combining results of a CT and ISCT, where the effective number of virtual patients, $n_0$, is dictated by the similarity of the results of the CT and the ISCT. The greater the similarity in the ISCT and CT results, the larger $n_0$, up to a pre-specified maximum. When $n_0 = 0$, only the clinical trial results are used to estimate the clinical parameter of interest. For larger $n_0$, the ISCT results are given greater weight, while still being constrained so that the CT results govern the overall result. All virtual patients are synthetic patients. Therefore, the ISCT and CT results are compared in distribution at the population level. However, the comparison is non-symmetric. For example, when the clinical parameter of interest is a failure rate, it is considered preferable for the ISCT to predict a greater failure rate than the CT, versus the other way around, and a non-symmetric metric for comparing the two distributions was proposed.

**2.3.7. Synthetic heart models for implantable cardioverter defibrillator evaluation.**
The next example of ISCTs is the work of [46] which uses large numbers of automata-based heart models for evaluating performance of implantable cardioverter defibrillators (ICDs). Unlike the example in Section 2.3.5 which involves a relatively small number (~20) of image-derived heart models, [46] develops a large number (> 11,000) of computationally inexpensive heart models. The baseline model is an automata-based model with a set of nodes and conduction paths connecting the nodes. Virtual patients are realized by prescribing electrophysiological timing and other parameters using uniform distributions with ranges obtained from the literature; in this manner > 11,000 models were generated covering 19 common arrhythmic conditions. The underlying algorithm and sensing method of two ICDs were implemented, and the virtual performance of the virtual ICDs was evaluated using the virtual cohort. The results matched a real clinical trial comparing the two ICDs, the Rhythm ID Going Head-to-Head Trial (RIGHT). In fact, the results of the original RIGHT trial were contrary to the trial hypothesis and therefore RIGHT was considered a failure; the fact that an ISCT was able to predict the same result demonstrates the potential value of ISCTs. A related work, [47], uses the same model within a Bayesian statistical framework for integrating the ISCT with a concurrent CT, using methods related to Section 2.3.6, and additionally presents a method for quantifying the robustness of the ISCT to underlying modeling assumptions.

**2.3.8 Fixation of leadless pacemakers.**   Leadless pacemakers improve upon traditional pacemaker systems by miniaturizing the pulse generator to such a degree that it can be implanted entirely within the heart, eliminating the need for leads to conduct energy to the tissue. Medtronic's leadless pacemaker, the Micra device, is delivered via a catheter and utilizes a novel fixation mechanism comprised of four nitinol tines. The tines are electrically inactive members that assure initial tissue to electrode contact, prior to expected encapsulation. Cyclic motion of the heart subjects these tines to a risk of fatigue fracture that could affect the initial contact. Fatigue related risks take time to manifest and thus require long patient follow-up times. Therefore, the fixation fatigue risk was addressed using a mixture of non-clinical data and modeling sources, including finite element modeling and Monte Carlo simulation. This approach incorporated variability in expected use conditions and material fatigue performance, allowing for a stochastic prediction of the risk of fixation fracture over long implant durations. This example is notable in that the "virtual patient" does not appear as a geometric representation of anatomy, but only through the description of the boundary conditions applied to the tines. This example is also notable because of the integration of finite element modeling with statistical modeling, generating a very large number of simulations. The study involved 5,000 simulated trials, each comprised of 10,000 virtual patients, for a total of 50 million simulations.

**2.3.9 Comparison of ISCTs.**   These eight examples showcase a range of possible attributes of ISCTs. Specifically:

- ISCT virtual cohorts could be comprised of subject-specific virtual patients such as in the MR-induced heating ISCT (Section 2.3.1), synthetic patients (see the ISCTs covered in Sections 2.3.2, 2.3.4–2.3.7), or a mixture of both types of virtual patient, such as FD-PASS (Section 2.3.3).

- Synthetic virtual patients are usually generated by sampling parameters from prescribed distributions. These distributions can be independent, such as the automata heart models (Section 2.3.7), or multivariate, such as the PCLC evaluation example (Section 2.3.4). However, other cases are possible, e.g., the whole heart models (Section 2.3.5) that are initially subject-specific but adapted to introduce pathological conditions so that they become synthetic models.

- The patient models may be systems of ODEs (such as in the PCLC evaluation example in Section 2.3.4), partial differential equations (such as the MR-induced heating example in Section 2.3.1 or the whole heart models in Section 2.3.5), rule-based (automata heart models in Section 2.3.7), multi-physics (VICTRE in Section 2.3.2 and FD-PASS in Section 2.3.3), or simply provide boundary condition values for the device model (Micra ISCT in Section 2.3.8).

- Device models may be 3D physics-based models (such as the MR-induced heating, VICTRE, FD-PASS or Micra ISCTs), implementation of device software algorithms (PCLC evaluation or automata heart models ISCTs), simple equations (Bayesian framework in Section 2.3.6) or simply provide boundary conditions for the patient model (whole heart ISCT in Section 2.3.5).

- ISCTs can be performed together with a real-world CT (Bayesian framework in Section 2.3.6 and [47] covered in Section 2.3.7); as a less expensive alternative to a full CT (VICTRE, whole heart model, automata heart model, and Micra ISCTs); to provide initial safety evidence prior to a CT (some PCLC evaluation use cases); or to address a question for which a CT would be unethical (MR-induced heating ISCT).

- Some ISCTs include a model of clinician behavior, such as the VICTRE ISCT.

## 3. Verification, validation and uncertainty quantification for ISCTs

We next consider verification, validation and uncertainty quantification. Our aim in this section is to highlight unique considerations for how each applies to ISCTs.

### 3.1. Verification

First, we consider verification, which can be broken down into two sub-activities: code verification, the process of identifying errors in the numerical algorithms of a computer code, and calculation verification, the process of determining the solution accuracy of a calculation [1].

For the most part, code verification considerations are the same for ISCTs as for other models. ISCTs are comprised of various sub-models and each need to be tested to ensure they have been correctly implemented. Errors in any of the sub-models could compromise the reliability of the ISCT, and it is important that unit testing is performed for all sub-models (including, for example, any clinical outcome model, or sampling method). Studies where multiple modeling groups attempt to solve the same pre-specified problem have been useful in various fields for code verification and, more generally, for scientific reproducibility [48–50]. A future "grand challenge" where multiple modeling groups independently perform a carefully specified ISCT could offer similar benefits to this emerging field.

In our previous work for patient-specific models we identified unique calculation verification considerations that arise for PSMs [9]. Some of these considerations also apply to ISCTs. First, we noted that with PSMs it is common to perform calculation verification studies such as grid refinement studies using a single, assumed representative, patient. For PSMs that are clinical tools, which can be applied to any new patient in the indicated patient population, this leaves open the possibility that the numerical error will be unexpectedly high for some patients. For example, suppose the mesh convergence study using one or a few patients determined that the spatial discretization size $h = h_0$ was sufficient for <1% error in the model output. If there are some patients in the indicated patient population for which errors with using $h = h_0$ are much greater than 1%, model predictions may be unreliable for those patients. Similarly, for ISCTs, it may be important to execute discretization error studies using multiple virtual patients from the cohort, ideally representing the entire virtual cohort, to develop confidence that the selected numerical parameters are appropriate for all virtual patients in the cohort. Strategies include (in no specific order):

- Compute the discretization error using two randomly sampled subsets of the virtual cohort and compare the consistency of discretization error.

- Compute the discretization error using a randomly sampled subset of the virtual cohort, estimate the 95% or 99% confidence interval for the discretization error, and use this information when analyzing virtual patient results in the ISCT.

- If computational time is a limiting factor, attention could be focused on expected worst-case virtual patients. For example, patients with anticipated large gradients of the primary quantities of interest that need a finer mesh to resolve.

User variability was also identified in [9] as a potential source of error for PSMs. Some PSMs include a manual step with a subjective decision in the model development workflow; for example, choosing manual segmentation or choosing a seed point for image segmentation. For such PSMs, it is important to confirm that outputs of the model are not impacted by inter- or intra-user variability [9,48,51,52]. The same principle applies to ISCTs that require subjective decisions in the creation of virtual patients.

## 3.2. Validation

What does validation of an ISCT look like? Unlike models of medical devices in isolation, or models of single patients, there are many features to an ISCT, and validation is possible at many levels. One possibility is to validate each of the different sub-models of an ISCT. Another possibility is to perform 'cohort validation', involving simulations from all members of the virtual cohort. Both cases are discussed below. Table 2 lists potential validation tests that arise in the subsequent sections.

**Table 2. Selection of possible validation tests that could be performed to evaluate an ISCT, identified in Section 3.2.** This is not intended to be a comprehensive list of options; many options or variants are possible.

| Test aim | Fig | Model | Comparator | Examples | Evidence Category in [2] |
|---|---|---|---|---|---|
| Validate device model | Fig 2a | Model of device physics | Bench test data using the real device | MR heating [28, 53], stent structural model [54], heart valve fluid-structure interaction [55] | 3 |
| Validate patient model | Fig 2b | Patient model using nominal or population-averaged parameter values | Population-averaged clinical data | Cardiac modeling [56] | 4 |
| Validate patient model | Fib 2c | A selected subject-specific virtual patient | Data (not used in model development) from the subject the virtual patient represents | Whole heart [57], automata-based heart [58], heart valve fluid-structure interaction [59] | 4 |
| Validate patient model | | Patient model based on some subject's data | Bench test constructed using the same subject's data | FD-PASS model [60] | 3 |
| Validate patient model | | "Patient model" personalized using cadaver derived data | Data (not used in model development) from the same cadaver | [61] | 3 |
| Validate coupled patient-device model | Fig 2d | A subject-specific virtual patient coupled to the device model | Data (not used in model development) from the subject the virtual patient represents, post-intervention | Stent-graft deployment [62] | 4 |
| Validate coupled patient-device model | | "Patient model" personalized using cadaver derived data, coupled to device model | Data (not used in model development) from the same cadaver after device deployed on cadaver | We are not aware of any examples. | 3 |
| Evaluate emulator | | Emulator | Full patient model | See [63] for a review of emulators for finite element based models. | N/A |
| Validate clinical outcome model | | Clinical outcome model | Independent dataset that was not used to create the clinical outcome model | We are not aware on any examples. [12] provides an example of developing a clinical outcome model and discusses challenges validating it. | N/A |
| Validate clinician model | | Clinician model | Independent dataset that was not used to create the clinician model | We are not aware of any examples. | N/A |
| Validate virtual cohort of subject-specific virtual patients | Fig 3a | All virtual patients in virtual cohort | Data (not used in model development) from each real subject, with individual-level comparison | Whole heart [64] | 4 |
| Validate virtual cohort | Fig 3b | Distribution of model output using all members of virtual cohort | Distribution of same quantity across the population using a clinical dataset | Whole heart [45] | 5 |
| Validate virtual cohort, validate ISCT methodology | Fig 3c, case 1 | Virtual cohort with device models, simulations to reproduce a previously performed clinical study | Previously performed clinical trial | VICTRE [32], FD-PASS [16], automata-based heart [46], validation using early feasibility study data [12] | 5 |
| Validate ISCT as part of ISCT/CT integration framework | Fig 3c, case 2 | Full ISCT results | Clinical trial results | Bayesian framework [26] | N/A |

**3.2.1 Sub-model validation.** One approach to ISCT validation is to validate each component of an ISCT separately and rely on this for overall credibility. In fact, if cohort validation is not possible, this may be the only approach to establishing ISCT credibility.

Considering first **device model** validation, it may be possible to perform validation of the device model in isolation, i.e., independent of the patient model (Fig 2a). For example, for MR-induced heating (Section 2.3.1), a model of the power absorption of a new device can be tested by comparing against experiments using the same device in a saline-gel phantom [53]. This evaluates if the new device can be modeled reliably. MR-induced heating ISCTs using the virtual patients such as the Virtual Population [30] often focus validation efforts on the device model (e.g., clause 8.7.5 in ISO/TS 10974 [28]), rather than the patient model or coupled device-patient model. This is because the virtual patients have been evaluated in previous studies and have been used in multiple studies; for this case, previously generated results essentially support the reliability of the patient model.

In some cases, no physics are simulated in the device model. Examples include some PCLC ISCTs (Section 2.3.4) where the device control algorithm is coupled to the patient model, and the heart model ISCT described in Section 2.3.5 where the CRT lead locations are prescribed but the device itself was not otherwise modeled. In such cases, traditional model validation methods are not applicable, but it will still be important to confirm device algorithms have been implemented correctly or device-related parameters are accurate.

Considering next the **patient model**, initially these will likely also be validated independently of the device model. Patient model validation may be one of the more challenging steps to demonstrating credibility of the ISCT. First, a decision is needed on what realizations of the baseline patient model to validate. If subject-specific virtual patients are generated (whether for cohort generation approaches A, B or C in Fig 1), validation may be possible against other data collected from the same subjects–see Fig 2b. (Note: for cohort generation approach A this directly validates the virtual patients in the cohort; for approaches B and C it validates models that form the basis of the virtual patients in the cohort). Alternatively, a synthetic patient representing an 'average patient' could be generated using nominal or population-average parameter values and then compared to population-averaged clinical data (Fig 2c; this assumes the 'average patient' yields average results, which may not be true) or experiments using, for example, a biomimetic phantom. Another option is creating a patient model using selected patient data and comparing against a bench test constructed using the same patient data. For example, [60] validates an aneurysm flow model against flow measurements in a corresponding glass model of the aneurysm and surrounding vasculature. A further option is constructing a 'subject-specific' model using data from a cadaver and validating against other data from the same cadaver.

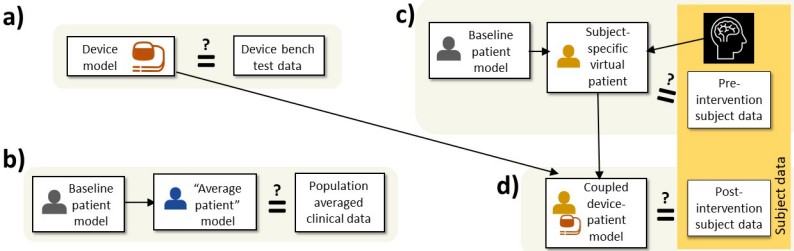

**Fig 2. Possibilities for validation of device and patient models (not comprehensive).** Equality sign with question mark indicates "Compared as part of validation activity". See text for discussion.

Next, it may be important to validate **coupled device-patient models**. For subject-specific virtual patients, it may be possible to collect pre- and post-intervention data from the same subject, enabling validation of both the virtual patient in isolation and the coupled device-patient model (Fig 2d; see [56] for an example in stent-graft deployment), or collecting pre- and post-interventional data using a cadaveric specimen. As an example, in the context of PCLC evaluation, a hemodynamic patient model could be integrated with a fluid resuscitation control system, consisting of PCLC algorithm and models for sensors, infusion pumps, and patient monitors. The coupled model could be tested through simulation of diverse clinical scenarios to confirm that the modeled patient and the system functions seamlessly and reproduces expected behavior across various operating conditions.

A key question is: do the virtual patients in the **virtual cohort** sufficiently cover variation in the population for the question of interest? None of the validation tests discussed above address this question. If it is not possible to do cohort-level validation (discussed below Section 3.2.2 and 3.2.3 below), it will be especially important to provide a justification for distributions used or to compare characteristics of the virtual cohort with the patient population.

Finally, it is important to validate any other sub-models in the ISCT, such as a clinician model, clinical outcome model, or models of other sub-systems. Results from an ISCT may only be as strong as their weakest link–for example, an ISCT with strongly validated patient and device models but an unvalidated clinical outcome model may not be reliable. However, it depends on the sensitivity of the ISCT results to those sub-models, and therefore sensitivity analyses could provide information on where to focus validation efforts. When evaluating sub-models that use data-driven or statistical methods (e.g., a statistical clinical outcome model), established cross-validation, bootstrapping and external validation techniques can be used [65, 66]. Developing and validating the clinical outcome model may be one of the greatest challenges to demonstrating credibility of an ISCT, due to the difficulty of collecting relevant data. For example, [12] proposes a method for performing an ISCT in parallel with a real-trial CT, where results from the CT are used to generate the clinical outcome model. The lack of independent data to validate the clinical outcome model is discussed.

Next, we consider validation involving *all* virtual patients in the finalized virtual cohort.

**3.2.2 Cohort validation—Individual-level comparison (paired data).**   For a virtual cohort made up of subject-specific virtual patients, it may be possible to perform individual-level validation using each cohort member. This could be a very powerful method of validating a virtual cohort to be used in an ISCT and involves the following workflow, as illustrated in Fig 3a: (i) collect clinical data from $N$ subjects; (ii) split each subject's data into model generation data and validation data; (iii) generate $N$ subject-specific virtual patients using the model generation data; (iv) validate each of the virtual patients using that patient's validation dataset. As an example, [64] constructs seven patient-specific heart (left atrium) models using electroanatomical data. During the procedure two pacing protocols were applied for each patient. One protocol was then used to calibrate the patient model and the second protocol used to validate each model.

As indicated in Fig 3a, the nature of the patient model output can dictate the analysis to be performed. When the patient models predict binary quantities (such as treatment success or presence of disease), accuracy of the virtual cohort could be quantified by computing sensitivities, specificities, area under receiver operating characteristics curve (AUC), etc., [67]. When the patient models predict scalar quantities on a continuous scale (such as temperature rise, displacement magnitude or peak stress), correlation or Bland-Altmann plots [68] may be appropriate. Other methods may instead be appropriate, depending on the application and data type.

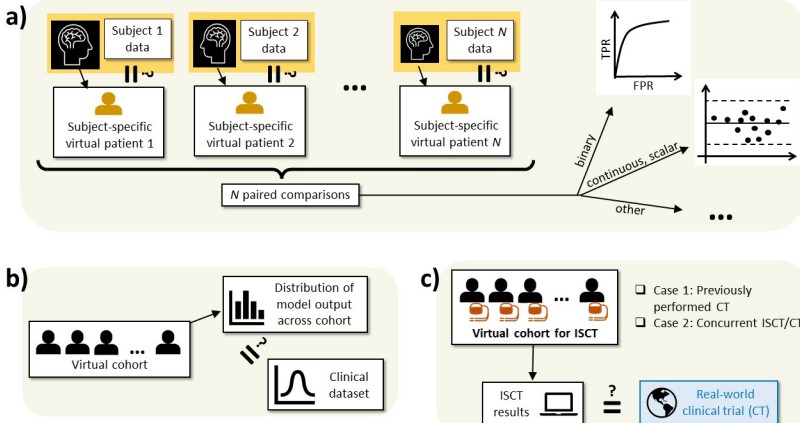

**Fig 3. Possibilities for cohort validation (not comprehensive).** Equality sign with question mark indicates "Compared as part of validation activity". (a) N subject-specific virtual patients each individually validated against data from the corresponding real-world subjects. (b) Distribution of a model output across the virtual cohort validated against corresponding data from the real-world population. (c) Results of an ISCT directly compared to a real-world clinical trial.

Individual-level validation will not be possible for most ISCTs, either because the virtual cohort has synthetic patients or because additional validation data was not collected (i.e., because all the data was used to calibrate the subject-specific virtual patients; nothing was withheld). In this case a population-level approach is the only option, discussed next.

**3.2.3 Cohort validation—Population-level comparison (no paired data).**   The other option for cohort validation is to compare a quantity that integrates outputs from all members of the cohort, against data from the population that the virtual cohort represents. Possibilities include:

- For one or more patient model outputs, computing the mean, variance, range and/or full distribution of a quantity across the virtual cohort, and comparing against corresponding data from a clinical dataset–see Fig 3b. For example, using several metrics quantifying cardiac rhythm management, [45] compared 95% confidence intervals obtained using a 24 subject virtual cohort with a 17 subject clinical study, mostly observing that the virtual cohort confidence intervals lay within the clinical confidence intervals. The advantage of this approach is that new independent validation of a virtual cohort can be performed as new clinical data becomes available, although there is a possibility of bias (e.g., cherry-picking datasets/evidence suppression). For the case of multiple model outputs, a stronger validation includes confirming if the model reproduces co-variances observed in the population.

- Reproducing the outcome of a specific medical device clinical trial–see Fig 3c. For example, in the VICTRE trial (Section 2.3.2) the endpoint was change in area under receiver operating characteristic curve (ΔAUC) between digital mammography and digital breast tomosynthesis. Means and standards deviations of ΔAUC were compared with results from an analogous CT. As another example, Section 2.3.7 describes how outcomes using a virtual cohort reproduced the results of the RIGHT trial comparing two ICD devices.

- Another possibility is the Bayesian frameworks of [26] (Section 2.3.6) and [47] (Section 2.3.7) in which the ISCT is envisaged to run in parallel with a CT (also see Fig 3c). This case differs from the other cases because the validation is arguably not part of a sequential

develop-validate-execute model workflow. Instead, the level of agreement between ISCT and CT (cohort validation results) directly determines the number of virtual patients used to augment the CT.

With population-level validation, discrepancies between virtual cohort outputs and clinical data could present as: different means, less or greater variability in the virtual cohort results, different shapes of distributions, or the presence or absence of observed co-variations. Next, we consider potential reasons for each of these, and implications.

- **Different means**: This could be due to deficiencies in the model, but it could also be caused by small sample size in either the virtual cohort or clinical comparator dataset. Note that if a virtual cohort is constructed using data from one single-center clinical study and then validated by comparing to a second single-center clinical study, the results should not be expected to agree any more than two single-center clinical studies. Also note that for some planned ISCTs, it could be argued that virtual patients conservatively over-estimating a mean quantity is acceptable, or at least preferable to under-estimating. For example, in [26] (discussed in Section 2.3.6) it is considered preferable for average failure rate from the virtual cohort to be greater than from a CT. To account for this, a non-symmetric validation metric for comparing the two distributions was proposed.

- **Less variability in virtual cohort**: This is a common occurrence (for an example in cardiac modeling, see validation results in supplement of [45]) and could be due to underestimation of the variability in specific parameters used to construct the virtual cohort–that is, the virtual patients do not adequately represent the real population. However, it could also be due to observational error in the clinical study. When a virtual cohort and clinical data agree in mean but the virtual cohort is less variable, it could be argued that the virtual cohort is potentially suitable for trial design but not trial replacement, although of course this depends on various other factors.

- **Greater variability in virtual cohort**: This could occur with virtual cohorts of synthetic patients sampled from distributions, where correlations between parameters is not accounted for and therefore non-physiological parameter combinations are sampled. It could be argued that for certain questions of interest this is acceptable, such as questions relating to safety concerns. For example, if the goal is to demonstrate device performance within safety limits across a wide range of virtual patients, it will be important to sample extremes of the population and some unrealistic patients may be acceptable.

- **Different shapes of distributions**: The case of similar levels of variability in the virtual cohort and the clinical data, but different distribution shapes (e.g., Gaussian for the model predictions but bimodal for the clinical data) immediately raises questions about the similarity of the virtual cohort to the clinical dataset participants, although the difference could also be due to the model structure and absence or lack of nonlinear relationships. Identifying the root cause of this difference may be challenging.

- **Differences in output co-variation**: This could be due to the model structure, which may fail to include an important mechanism. For example, women are known to have on average smaller hearts and have differences in electrophysiology [69, 70], and this relationship between heart size and tissue properties may not be explicitly represented in a model. The absence or presence of an unexpected co-variation in model outputs may indicate that an important mechanism is missing in the model. However, the number of covariances increases quadratically with the number of outputs, and collecting multiple outputs can be challenging, meaning covariance validation may be hard to achieve in early virtual patient cohorts.

Conversely, it is worth considering the level of credibility imparted if the virtual cohort and clinical data do agree in both mean and variability, and the potential danger of relying on population-level validation alone, especially if a virtual cohort was generated by matching population-level data for a similar cohort. ISCTs are hierarchical models built from many components, and only validating an integrated top-level quantity presents the risk of getting the "right answer for the wrong reason" [71]. Accordingly, we believe population-level validation should support sub-model validation, rather than replace it.

Finally, we note that for very large virtual cohorts, it may be desirable to use a random sample of virtual patients from the cohort, rather than all of them, in cohort validation. It should be clear from this section that there are numerous potential strategies to ISCT validation, and many variants or alternative approaches are possible. Table 2 should in no way be considered comprehensive.

### 3.3. Uncertainty quantification

UQ in medical device applications is generally considered difficult. Therefore, comprehensive UQ for ISCTs will likely be considered challenging in the near term. However, one of the UQ challenges for models with medical device applications is quantifying the potential impact of population variability, and ISCTs explicitly address population variability through the virtual cohort, so this does not need to be addressed through separate UQ. Still, there are other sources of uncertainty that could impact predictions, including in (but not limited to):

- device model parameters,

- the personalized parameters for subject-specific virtual patients due to measurement error,

- virtual patient fixed parameters,

- any assumed probability distributions,

- other ISCT sub-models, such as variability in clinical treatment protocols, and

- validation comparator data, e.g., observation error in clinical data that will be used to validate a virtual cohort.

The last of these is especially consequential, since, as discussed in Section 3.2.3, if greater variability is observed in the clinical dataset than the virtual cohort, it can be difficult to know if this is due to limitations of the modeling or of the clinical data.

For the other sources of uncertainty, identifying which are most consequential is another challenge. A decision on where to focus UQ efforts could be based on expert judgement or on sensitivity analysis results. Note that the Bayesian framework of [26], discussed in Section 2.3.6, does formally address uncertainty in device parameters.

Since model outputs can be strongly affected by underlying uncertainties in a biomedical model, it could be beneficial when planning a UQ analysis to demonstrate that the ISCT study *conclusion* is not impacted by the uncertainty, rather than focusing on the model output alone. As an example, in [9] we performed UQ for a cardiac virtual cohort simulation study and demonstrated that the study conclusion is not affected by uncertainty in personalized anatomic regions.

ISCTs are comprised of sub-models, so a complete UQ effort could attempt to quantify the total prediction uncertainty based on quantification of the uncertainty of the various sub-models. Methods for performing such a bottom-up analysis are available in the UQ literature [72–74]. The uncertainty in the final predicted outcome can be assessed using techniques such as sampling (including Monte Carlo sampling) or polynomial chaos expansion. Once again,

the technical difficulty in performing such an analysis should not be underestimated. It is also worth noting that rigorous bottom-up UQ is not generally performed as part of clinical trials. For example, propagation of the uncertainty of the clinical measurement system (e.g., blood hemoglobin measurement) to the final trial results is not common. As ISCT practice evolves, there needs to be consideration as to whether UQ will make ISCT results appear to be of lower quality (i.e., higher uncertainty) when compared to traditional human trials because of the additional rigor.

## 4. Workflow for Assessing Credibility of a Medical Device ISCT

There is a need for standardized frameworks and workflows for evaluating computational models with medical device applications to ensure rigor, transparency, and uniformity in how the models are assessed. ASME V&V40-2018 [1] began to address this need, by providing a credibility assessment framework based on the risk associated with using the model and so-called 'credibility factors' that assess the rigor of the VVUQ activities. ASME V&V40 defines 23 credibility factors to assess the rigor of the VVUQ activities, related to code verification, calculation verification, the model, the validation comparator, and applicability of the validation activities to the COU. For example, two code verification credibility factors are defined: 'software quality assurance' and 'numerical code verification'. To follow ASME V&V40, users should define a gradation of activities for each factor, corresponding to increasing levels of investigative rigor, and select a level of rigor based on the assessed model risk. ASME V&V40 provides example gradations for each factor. See ASME V&V40 [1] for the full list of credibility factors and their gradations.

Challenges with ASME V&V40 for ISCTs include the assumption in ASME V&V40 that there will be one validation comparator only and language geared towards bench test validation. The FDA guidance "Assessing the Credibility of Computational Modeling and Simulation in Medical Device Submissions" [2] provides a more general framework, based on ASME V&V40. It makes no assumption about the type of validation comparator, and defines eight categories of credibility evidence, including three types of validation: bench test validation, *in vivo* validation (validation against animal/clinical data with individual-level comparison), and population-based validation, which corresponds to the validation activities described in Section 3.2.3. Following the framework in [2], users define credibility factors and gradations appropriate for their planned credibility evidence. The guidance [2] is not prescriptive on all technical implementation steps to ensure it is relevant to a wide range of models and modeling applications. Therefore, there is a need for example workflows for using the guidance framework, including for ISCTs, to accelerate the field of credibility assessment. Here, we consider one example of how the framework in [2] could be applied for ISCTs.

It is clear from Section 3 that ISCT validation can be hierarchical, and that there are many options for validating sub-models within an ISCT, as shown in Table 2. This means that it is difficult to apply ASME V&V40 directly to ISCTs, but the flexibility of the framework in [2] makes it more suitable for ISCTs. However, given the wide range of possibilities for ISCTs, we do not attempt to propose an ISCT credibility assessment framework that covers *all* possible validation strategies. Instead, we provide in this section an *example* workflow relevant for some cases, that could be adapted as needed depending on the sub-models in the ISCT and the validation strategy. The example workflow presented below takes a hierarchical approach to validation, validating device sub-model against bench data (Fig 2a), patient sub-model and coupled device-patient model using *in vivo* patient data (Fig 2b and 2d), and the virtual cohort against population-level data (Fig 3b). The workflow is fully aligned with the framework in [2]. Initial Steps (Steps 1–3: State question of interest, describe context of use, and perform risk

assessment) and latter Steps (Steps 6–8, related to adequacy assessment and execution of credibility assessment studies) simply refer to [2]. Step 4 describes specific step-by-step VVUQ activities to perform to evaluate the device, patient, coupled device-patient and virtual cohort sub-models. Step 5 discusses specific credibility factors and gradations that could be used, including a novel set of credibility factors and gradations for population-based validation of the virtual cohort. This ISCT workflow is a sister workflow to a previous workflow we developed for evaluating patient-specific models in medical device software [75], that was based on the analysis in the sister article [9].

The workflow assumes the following:

- An ISCT will be performed to address a safety or effectiveness question for a medical device.

- The ISCT will use a physics-based device sub-model.

- The ISCT will use a first principles-based patient model requiring the solution of ODEs or PDEs.

- There is no emulator or explicit clinician model

- The ISCT will use a data-driven clinical outcome model to convert patient model outputs to clinical outcomes.

- Clinical data are available from a small number of subjects for patient model validation. Each subject's dataset contains: (i) data that can be used to construct a subject-specific virtual patient; (ii) pre-intervention data that can be used to validate the subject-specific virtual patient; and (iii) post-intervention data that can be used to validate the coupled device-patient model.

- The ISCT will use a virtual cohort of synthetic patients generated through sampling, for example sampling probability distributions based on literature data or other relevant data.

- The ISCT will not use a Bayesian framework to integrate ISCT results with a real-world clinical trial.

The workflow is intended as a starting point for developing an ISCT credibility assessment plan and can be modified based on the specific ISCT details (such as specific sub-models and protocol), and feasibility of collecting relevant data. Possible modifications to the workflow for cases different to the above are discussed in S1 Text.

A hypothetical whole heart model ISCT will be used to illustrate some of the sub-steps in Step 4 below. Suppose a virtual cohort of heart models will be used to assess durability of a cardiac pacemaker lead under realistic loading conditions with specific attention to lead motion in the vicinity of the heart. The device model is a model of the lead mechanics for an implantable pacemaker. The patient model is a model of the deformation of the heart during the cardiac cycle. A virtual cohort of synthetic heart models is generated by sampling probabilistic representations of heart geometry across a patient population, and will be used in an ISCT to evaluate lead durability. Below, we will consider how this hypothetical ISCT could be evaluated. Note that the example is entirely hypothetical and intentionally limited in detail to keep it high-level. It is intended to illuminate the steps below only; we make no claims about the feasibility or value of such an ISCT, or the feasibility or value of the specific credibility tasks chosen.

Given the above assumptions, one possible workflow for ISCT credibility assessment, following the general framework in [2] is:

1. State the Question of Interest. See [2] for specific recommendations.

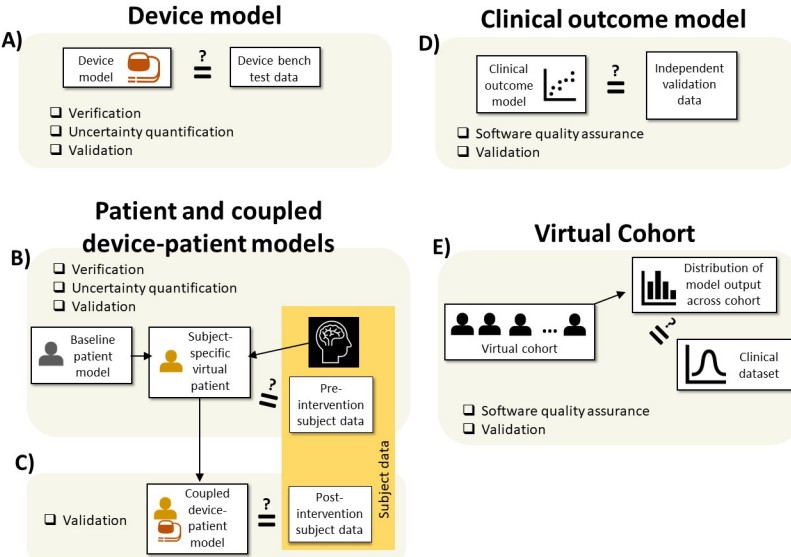

**Fig 4. Overview of sub-steps in Step 4 of the workflow.** Equality sign with question mark indicates "Compared as part of validation activity".

2. Describe the Context of Use. See [2] for specific recommendations.

3. Assess Model Risk. See [2] for specific recommendations.

4. Plan the following activities to generate model credibility evidence–see Fig 4 for an overview of all activities in this step. 'Categories' in the below refer to the categorization of credibility evidence in [2]. Bolded terms are activities to be performed. Examples under each step refer to activities that could be performed for our hypothetical ISCT described above.

A. Assess the device sub-model in isolation

 a. Code verification results (Category 1)

 i. **Software quality assurance**. Refer to ASME V&V40 [1].

 • e.g., software quality assurance of the lead mechanics code.

 ii.**Numerical code verification**. Refer to ASME V&V40 [1].

 • e.g., use the method of manufactured solutions to verify the lead mechanics code.

 b. Bench test validation results with supporting calculation verification and UQ (Category 3)

 i. **Bench test experiments:**
 Perform device bench testing to generate comparator data for device model validation. See Section 3.2.1 for examples.

• e.g., perform mechanical bench experiments on selected leads.

 ii. **Discretization error analysis**:
 Using the uncoupled device model that will be used in the device model validation simulations, perform a mesh convergence analysis to estimate numerical error. If time-dependent governing equations are solved, also perform a time-step convergence analysis. Refer to ASME V&V40 [1].

- e.g., set up lead mechanics simulations replicating the bench test, and then perform a mesh resolution convergence analysis.

  iii. **Numerical solver tolerance analysis:**
   Using the uncoupled device model that will be used in the device model validation simulations, perform a sensitivity analysis in the numerical solver tolerance(s). Refer to ASME V&V40 [1].

  - e.g., confirm that the linear solver tolerance in the lead mechanics solver is sufficiently small to not impact the solution.

  iv. **Use error assessment:**
   Perform use error assessment to confirm that no errors were made in the device model setup or validation simulations. Refer to ASME V&V40 [1].

  v. **Uncertainty quantification:**
   See Section 3.3 and ASME V&V40 [1].

  - e.g., identify key parameters (perhaps through sensitivity analysis; possible key parameters include lead material properties) and assess the impact of uncertainty in these parameters on simulation outputs.

  vi. **Validation:**
   Compare the bench test experimental results with the device model validation simulations. See Section 3.2.1 and ASME V&V40 [1] for discussion.

B. <u>Assess the patient sub-model in isolation</u>

  a. Code verification results (Category 1)

  i. **Software quality assurance**: Refer to ASME V&V40 [1].

  - e.g., software quality assurance of the cardiac mechanics code.

  ii. **Numerical code verification**: Refer to ASME V&V40 [1].

  - e.g., compare results of the cardiac mechanics code with other cardiac mechanics solvers using benchmark test problems.

  b. *In vivo* validation results with supporting calculation verification and UQ (Category 4)

  i. **Identify or generate a clinical validation dataset**:
   Identify a dataset for patient model validation. Each subject's data will be split into model personalization data, observations taken pre-intervention, and observations taken post-intervention, as shown in Fig 4.

- e.g., collect pre- and post-operative imaging data for a representative patient, or small number of patients, implanted with a pacemaker, including MR, cine-MR and fluoroscopy imaging data.

  ii. **Generate models**:
   Using each subject's model personalization data, generate subject-specific virtual patients.

  - e.g., generate subject-specific heart models from the pre-operative imaging data.

  iii. **Discretization error analysis**
   Using a subset (or all) of the subject-specific virtual patients, perform a mesh

convergence analysis to estimate numerical error. If time-dependent governing equations are solved, also perform a time-step convergence analysis. See Section 3.1 for discussion.

- e.g., perform mesh convergence analysis using a subset of the above heart models.

iv. **Numerical solver tolerance analysis:**
Using a subset (or all) of the subject-specific virtual patients, perform a sensitivity analysis in the numerical solver tolerance(s). See Section 3.1 for discussion.

- e.g., confirm that nonlinear and linear solvers tolerances in the cardiac mechanics solver are sufficiently small to not impact the solution.

v. **Use error assessment:**
Perform use error assessment to confirm that no errors were made in the patient model setup or validation simulations. Refer to ASME V&V40 [1].

vi. **Uncertainty quantification:**
See Section 3.3 and ASME V&V40 [1] for discussion. Note that patient model UQ may not be feasible due to lack of data or computational cost, in which case a justification for not performing UQ could be provided instead, e.g., based on sensitivity analysis.

vii. **Validation:**
Compare each patient model output(s) with that subject's pre-intervention data. See Section 3.2.1 for examples and discussion.

- e.g., compare predictions of the deformation of the heart vs actual motion observed from cine-MR imaging data for each patient.

C. Assess the coupled device-patient model

a. *In vivo* validation results (Category 4)

i. **Generate coupled models:**
For each of the subject-specific virtual patients created in 4B above, couple with the device model to create coupled device-patient models, using the same mesh resolution, solver tolerance and other factors finalized in earlier steps.

- e.g., generate coupled pacemaker-heart models for the heart models generated in 4B.

ii. **Validation:**
Compare each coupled device-patient model output with that subject's post-intervention data. See Section 3.2.1 for discussion.

- e.g., predict the motion of the lead in the coupled pacemaker-heart models during the cardiac cycle and compare with observed motion from the fluoroscopy imaging data.

D. Assess the clinical outcome model
Assuming the clinical outcome model is a data-driven model, evaluate the credibility of this model using appropriate methods (e.g., statistical model validation methods as discussed in Section 3.2.1). It is also important to perform software quality assurance to ensure the clinical outcome model is functioning correctly.

E. Assess the virtual cohort

a. Code verification results (Category 1)

  i. **Software quality assurance**. Refer to ASME V&V40 [1].

   • e.g., software quality assurance of cohort generation code.

 b. Population validation results (Category 5)

  i. **Identify population-level clinical dataset:**
  Identify a dataset providing information on the mean, variance, distribution or other statistical metrics, of a relevant quantity(ies) across the population. See Section 3.2.3 for discussion.

   • e.g., collect statistics on cardiac biomechanics (e.g., ejection fraction, peak global longitudinal strain) for the relevant patient population.

  ii. **Generate the virtual cohort and compute corresponding model outputs:**
  Generate the virtual cohort of synthetic patients and compute the same quantity(ies) for each synthetic patient.

   • e.g., compute the distribution of the same cardiac biomechanical parameters using the virtual cohort.

  iii. **Validation:**
  Compare the results from the virtual cohort with the clinical dataset using an appropriate method of comparison. See Section 3.2.3 for discussion including implications of different possible results.

   • e.g., compare the statistics of cardiac biomechanical parameters from the virtual cohort to that in the patient population.

5. Define credibility factors and gradations for the planned credibility evidence, and evaluate the planned activities using the gradations, as discussed in Step 5 of the Guidance.

A. <u>Device sub-model in isolation</u>
One option is to use the credibility factors and gradations in ASME V&V40 [1].

B. <u>Patient sub-model in isolation</u>
One option is to use the credibility factors and gradations for patient-specific models (e.g., [75]), adapted as appropriate.

C. <u>Coupled-device patient model</u>
One option is to use the validation credibility factors and gradations from 5B.

D. <u>Clinical outcome model</u>
Credibility factors are not applicable to the clinical outcome model in this example, because the clinical outcome model was assumed to be a data-driven model, which is outside the scope of the FDA Guidance [2] and ASME V&V40 [1].

E. <u>Virtual cohort</u>
Table 3 lists credibility factors and gradations that could be used or adapted for the population-level validation.

6. Assess Adequacy of the planned activities–see Guidance Step 6 for details.

7. Execute studies.

8. Assess Adequacy of the overall results–see Guidance Step 8 for details.

**Table 3. Credibility factors and gradations that could be used for population-level validation of the virtual cohort.**

| | Credibility factor | Credibility gradation |
|---|---|---|
| Clinical dataset | Number of subjects in clinical dataset | (a) Small number of subjects<br>(b) Multiple subjects, not enough to be statistically relevant<br>(c) Statistically relevant number of subjects |
| | Range of characteristics of subjects in clinical dataset | (a) Subjects used to validate virtual cohort represent a very narrow range of characteristics (e.g., average) of the intended population.<br>(b) Subjects used to validate virtual cohort represent a range of characteristics of the intended population, but not the entire range.<br>(c) Subjects used to validate virtual cohort represent the entire range of characteristics of the intended population. |
| | Patient-level data available | (a) Key patient data missing<br>(b) Most key patient data are available.<br>(c) All key patient data are available. |
| | Source of data | (a) Data from a single center study<br>(b) Data from a multi-center study or multiple studies |
| Comparison | Output comparison–quantity | (a) A single output was compared.<br>(b) Multiple outputs were compared. |
| | Equivalency of output parameters | (a) Types of outputs were dissimilar.<br>(b) Types of outputs were similar.<br>(c) Types of outputs were equivalent. |
| | Rigor of output comparison | (a) Means were compared only<br>(b) Means and variance/standard deviation/range were compared<br>(c) Full distributions were compared |
| | Agreement of output comparison | (a) The level of agreement of the output comparison was not satisfactory for key comparisons.<br>(b) The level of agreement of the output comparison was satisfactory for key comparisons, but not all comparisons.<br>(c) The level of agreement of the output comparison was satisfactory for all comparisons. |
| Applicability | Relevance of the quantities of interest (QOIs) | (a) The QOIs computed in the population-level validation were not closely related to those to be used in the ISCT<br>(b) The QOIs computed in the population-level validation were closely related, though not identical, to those to be used in the ISCT<br>(c) The QOIs computed in the population-level validation were identical to those to be used in the ISCT |

## 5. Discussion

The purpose of this paper was to identify approaches and unique considerations for ISCT credibility assessment. We also aimed to provide a workflow that aligns with FDA guidance [2] for evaluating ISCTs. We highlighted how ISCTs are composed of multiple sub-models, including the device model, baseline patient model, coupled device-patient model, virtual cohort, and potentially other models such as a clinical outcome model, clinician model, and emulator. For this reason, a hierarchical approach to credibility assessment, where each sub-model is tested separately, is crucial. We identified a wide range of credibility assessment options for the sub-models, as illustrated in Table 2. Therefore, the workflow we proposed in Section 4 does not aim to cover every conceivable approach to ISCT validation but can be adjusted depending on case specifics.

The workflow we presented offers several advantages over *ad hoc* ISCT evaluation approaches. First, it is systematic, which can reduce planning costs, ensure uniform evaluation of ISCTs, and potentially alleviate stakeholder reliability concerns. Second, it is hierarchical,

ensuring individual testing of sub-components within the ISCT. This is crucial for ensuring that the overall ISCT model accurately predicts "the right answer for the right reasons" [71]. Third, it is based on the ASME V&V40 [1] approach, making it risk-based. The level of rigor in the credibility activities is selected based on the risk associated with using the model, limiting unnecessary evaluation efforts. Fourth, it aligns with the framework in the [2], promoting planning and discussion among stakeholders before initiating credibility assessment efforts. This alignment should optimize the process. However, there are some disadvantages to the workflow. It may be perceived as overly burdensome, and adjustments may be necessary based on the specifics of the ISCT, potentially introducing subjectivity into the process.

Overall, we anticipate that the general approach of the workflow will be applicable to many medical device ISCTs. We believe it should be possible to define a variant of the workflow suitable for each of the example ISCTs reviewed in Section 2.3, except for the Bayesian framework discussed in Section 2.3.6, which differs fundamentally from the others.

The scope of this paper was credibility assessment of first-principles based models used in ISCTs. A major challenge in evaluating ISCTs is that many ISCTs will likely combine first principles-based models with data-driven models such as machine learning models. Data-driven methods are especially likely to be employed for the clinical outcome model and emulator. These can be evaluated in isolation using standard methods for data-driven models. However, data-driven models may also be used for the device, patient, or certain aspects of these. Presently, credibility assessment for first-principles and data-driven models are not harmonized, which makes it challenging to consider when they are combined. A high-level holistic framework for assessing credibility of all computer-based models is needed to address this challenge.

Setting up and executing an ISCT may require a variety of simulations, including: (i) simulations for initializing patient models to a physiological state of interest; (ii) simulations to compute a quantity used in inclusion-exclusion criteria (e.g., for cardiac models, run each virtual patient to compute ejection fraction); (iii) simulations to deploy the medical device (e.g., solving contact problems to compute the exact position of an implantable device); and (iv) simulations to compute the final ISCT results. When multiple simulations are required, the applicability of the validation results to each of the simulations should be considered, to ensure there are no limitations in the credibility assessment activities.

ASME V&V40 2018 is a risk-informed framework, as is the FDA framework [2] that builds upon it. Decision consequence, which is the significance of adverse events (generally to patients) if an incorrect decision is made, impacts credibility requirements. It is worth considering what types of questions ISCTs may be used to address. Using an ISCT to answer a low consequence question may seem impractical, given the considerable effort involved in ISCTs. We expect that the greatest impact will come from higher-consequence questions. However, we have discussed how one application of ISCTs is to refine inclusion-exclusion criteria for a future real-world clinical trial. This arguably has lower patient decision consequence, although the business consequence could be significant, if it led to an unsuccessful trial or a narrower indicated patient population than necessary.

The scope of this paper was credibility assessment of an ISCT only; execution of the ISCT was outside the scope. Like credibility assessment, there are many options available for ISCT study execution, including options that might not be feasible in a real-world study, such as applying control and intervention to all virtual patients, or, in the case of multiple interacting interventions, combinatorically considering all possibilities for each virtual patient. Ad hoc removal of virtual patients, for example based on non-physiological behavior, is also a possibility in ISCTs. This range of options for executing an ISCT further complicates the development of standardized processes. Establishing good practice for all stages in an ISCT, of which the credibility assessment will be a subset, will be vital to the success of the field.

Large-scale acceptance of ISCTs will require more than rigorous ISCT credibility assessment. There are many challenges that will need to be overcome for ISCTs to be successful. One is related to terminology. Standardization is key to maturation of novel methods, but standardization is difficult without consistent terminology. Currently, there is no community accepted definition of an *in silico* clinical trial. We provided our interpretation in Section 2. Even with our definition, there is potential confusion as to whether cases such as Section 2.3.8 is one ISCT or 5,000 ISCTs. Additionally, the field may benefit from clear terminology for combined real-world clinical trials and ISCTs.

To make the greatest impact, a major challenge will be ensuring that ISCTs can provide the same information that would normally be obtained from a traditional clinical trial, including adverse events. Clinical trials are generally designed with the minimum number of patients that are required to provide adequate statistical power. Because of this, for any case for which an ISCT would reduce the patient count, it is imperative that ISCTs can provide the same information that would be obtained from a traditional trial, including adverse events, which are often a very important trial outcome. This implies that a simulation pipeline that was developed for a particular ISCT might not be directly re-usable for another ISCT with different endpoints, because basic assumptions in the initial ISCT (like the realism of the anatomical models) might bias the second one.

Related to credibility assessment are questions of data quality and transparency. Low quality data can lead to biased ISCT results if the data are not representative of the target population. Measurement errors can have a profound impact on model outputs. Opaque model development or parameterization practices can lead to challenges evaluating ISCT credibility or poor explainability of predictions. Success of the ISCT field will require comprehensive and diverse data collection, together with transparent working practices and the ability to understand and interpret model outputs and the internal mechanisms producing those outputs.

Clinical trial practice relies on a robust foundation of biostatistics. However, for ISCTs it may be necessary to develop new or adapted statistical methodologies to accurately assess device efficacy and safety. While this paper does not attempt to delve into this complex area, several considerations are noteworthy. These include integrating model uncertainty into the statistical analysis, accommodating the diverse range of possibilities inherent in ISCTs (such as administering both control and therapy for each virtual patient, or simulating multiple interacting therapies in a combinatorial manner), integrating ISCTs with conventional clinical trials (as discussed in [26]), and tailoring protocols specifically for ISCTs, given the distinct factors that constrain them compared to traditional clinical trials. As ISCTs continue to mature, we anticipate that the statistical theory underpinning them will emerge as a burgeoning field ripe for exploration.

Even more so than other applications of M&S in healthcare, there are cultural challenges that will need to be overcome for ISCTs to achieve success in reducing or refining clinical trials or animal testing. We believe that it will be critical that the direct stakeholders (industry and regulators) see tangible benefits of ISCTs and that there is clear communication of the reliability of ISCTs to other stakeholders (patients, clinicians, payers). Success stories with ISCTs will be important. One strategy to achieve these could be ISCTs using patient-specific models that are already approved for clinical use (e.g., using a patient-specific modeling tool that is already on the market as a medical device, to generate a virtual cohort for evaluating a new medical device).

Overall, there are many challenges that need to be addressed, and success of the field is by no means assured. However, we believe that formalizing and harmonizing the credibility assessment process, as advanced by this paper, will be a major step towards ISCTs achieving their potentially transformative impact on human healthcare.

## Disclaimer

The mention of commercial products, their sources, or their use in connection with material reported herein is not to be construed as either an actual or implied endorsement of such products by the Department of Health and Human Services. This article reflects the views of the authors and should not be construed to represent FDA's views or policies.

## Supporting information

**S1 Text. Supplementary material.** The supplementary material document discusses possible modifications to the example workflow in Section 4 for ISCTs that are different to the case considered in Section 4.
(PDF)

## Author Contributions

**Conceptualization:** Pras Pathmanathan.

**Investigation:** Pras Pathmanathan, Kenneth Aycock, Andreu Badal, Ramin Bighamian, Jeff Bodner, Brent A. Craven, Steven Niederer.

**Methodology:** Pras Pathmanathan, Kenneth Aycock, Andreu Badal, Ramin Bighamian, Jeff Bodner, Brent A. Craven, Steven Niederer.

**Project administration:** Pras Pathmanathan.

**Writing – original draft:** Pras Pathmanathan, Kenneth Aycock, Andreu Badal, Ramin Bighamian, Jeff Bodner, Brent A. Craven, Steven Niederer.

**Writing – review & editing:** Pras Pathmanathan, Kenneth Aycock, Andreu Badal, Ramin Bighamian, Jeff Bodner, Brent A. Craven, Steven Niederer.

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
