## [Decision Letter · Decision Letter 0]

1 Feb 2024

Dear Dr. Pathmanathan, 

Thank you very much for submitting your manuscript "Credibility Assessment of In Silico Clinical Trials for Medical Devices" for consideration at PLOS Computational Biology.

As with all papers reviewed by the journal, your manuscript was reviewed by members of the editorial board and by several independent reviewers. In light of the reviews (below this email), we would like to invite the resubmission of a significantly-revised version that takes into account the reviewers' comments.

We cannot make any decision about publication until we have seen the revised manuscript and your response to the reviewers' comments. Your revised manuscript is also likely to be sent to reviewers for further evaluation.

Sincerely,

Mukesh Dhamala

Guest Editor

PLOS Computational Biology

Ruth Baker

Section Editor

PLOS Computational Biology

Reviewer's Responses to Questions

**Comments to the Authors:**

Reviewer #1: This is a very interesting paper with a variety of real-world implications.

I think this needs to be clearly stated within the manuscript; there should be a clear statements made about the advantages and disadvantages of this method as well as its application in a real-world setting. What type of medical devices can be used here? there are many different categories and how is this aligned to the current regulatory landscape? simulation methods have been used for a variety of reasons and accepted in some situations. Hence, this should be stipulated within the context of the paper.

What do you mean about acute to long term outcomes? acute and chronic are very specific clinical terms and should not be used in this context as the details written in this section is not really what acute would mean. How would you determine short, medium and long term clinical outcomes? did you try this approach with a specific condition given the complexities linked to communicable and non-communicable diseases.

Reviewer #2: The manuscript provides a compelling exploration of in silico clinical trials (ISCTs) for medical devices, emphasizing in particular the crucial aspect of credibility assessment. This focus is highly relevant in today's context where computational models are increasingly integral to medical research.

As a biostatistician with primary experience in clinical trials, I find the topic of ISCTs both important and challenging. Although I have not worked directly on ISCTs, the manuscript provides insightful examples that illustrate the workflow of ISCTs, highlighting key areas such as modeling and simulation (M&S), verification and validation, and uncertainty quantification. These examples are particularly helpful in understanding the complexities and nuances of ISCT.

However, from a statistical standpoint, my ability to provide in-depth analysis is somewhat limited, as the paper serves primarily as a review and opinion piece rather than a research study with empirical data. It would be beneficial for readers if future work could include more detailed statistical analyses or case studies that demonstrate the practical implementation and challenges of ISCT in medical device development. Such contributions would be invaluable to statisticians and other researchers interested in exploring this emerging field.

In conclusion, the manuscript successfully highlights the importance and challenges of ISCTs in the current medical landscape. It serves as a foundational piece for further discussion and research in this evolving field, particularly for those of us in the biostatistics community looking to expand our understanding of ISCT methodologies.

Reviewer #3: Summary:

In this paper, the overall current state of ISCT model approaches and the respective selection/validation/etc. techniques are reviewed. The authors present different levels of validation and credibility assessment (patient, cohort, device, clinician, etc.), and how they are all interconnected. Specific examples of each type of sub-model are and how they can be used for augmenting or replacing clinical trials (in some cases) are provided. Through this review, several important factors are presented for consideration when creating and using an ISCT model, such as errors, variances, and distributions of the cohort(s).

The paper does a good job of presenting existing models and approaches and is successful in explaining the motivation and need for standardized approaches in modeling and simulation. I particularly liked the emphasis on the hierarchical nature of a simulated clinical trial; it was interesting to read about the "building blocks" (in a way) of a virtual cohort.

Main points of concern:

My main point of criticism concerns the overall focus of the manuscript and the manuscript’s presentation of the proposed credibility assessment workflow.

By "overall focus" I mean the following comments –

From my perspective, there is an overuse of “refer to this work” or “see [##]” type of sentences and it makes the manuscript seem less cohesive. Many of these instances can be supplemented with the key takeaway(s) from the referenced work, rather than directly referring the reader to other articles.

For instance, in section 2.3.4 (lines 308-309), consider the following statement: “PCLC evaluation provides an example of case C in Figure 1; see e.g., [37], which considers how to generate synthetic patients from calibrated subject-specific virtual patients.” This could be improved by more context regarding [37] and the citation, rather than the “see e.g., [37]” type of redirection. By this, I mean something more like: “PCLC evaluation provides an example of case C in Figure 1; one such example is through {explanation of the cardiovascular responses and methods presented in that conference paper} [37], which considers how to generate synthetic patients from calibrated subject-specific virtual patients.”

(My addition/suggestion is within the {} brackets above)

Or, in this statement: “Developing and validating the clinical outcome model may be one of the greatest challenges to demonstrating credibility of an ISCT, due to the difficulty of collecting relevant data; see [12] for a discussion.”

It would be nice to provide one or two key takeaways from that discussion in your text, so that the reader can have an idea without having to search for it.

Regarding my comment about the presentation of the proposed example workflow –

I feel as though there isn’t enough focus on it, since it is such a shorter section when compared to the rest of the text. I believe that this is one of the core components of this manuscript, but it feels a little lackluster in the main text. While I understand that there is more detail in the supplementary text, I think it would be beneficial to provide more information about its use. If possible, it could be interesting to present a specific example of this workflow being implemented. There are many general examples provided throughout the manuscript (which is a good thing), but the utility of the framework would be more clearly demonstrated if one potential specific implementation is described. For instance, if you presented a start-to-finish application (in an abstract manner) of this workflow regarding the study of a disease or device in a specific population and consider the models that would be needed for that, how they would be validated, etc. in reference to your workflow.

I understand that the execution of the ISCT itself is outside of the paper’s scope, but I feel like a little more elaboration in the main text would be useful for contextualization of the workflow.

Figures:

Figure 1 – I think it would be beneficial to also make the border of the “possible components” a dashed line. The light blue is difficult to view at times, so I think that also having a design difference (by changing the border) would be useful in visualizing the flowchart. Additionally, a brief description of the left pane should be included; only the right pane has a description currently.

Figure 3 – In the caption, there should be a brief description of a), b), and c) and their components, rather than or in addition to “See text discussion for discussion”.

Tables:

Table 2 – When providing the examples, I think it would be better to have the example and then the citation. For instance: “MR heating [28, 53], stent structural model [54], heart valve fluid-structure interaction”

Typos and minor comments:

1. Introduction, line 67: The statement “The aims are similar;” seems like it’s missing stuff, and the semicolon should be a colon.

2. Introduction, lines 70-72: “In this paper our focus is ISCTs for medical devices. However, our conclusions may inform evaluation of ISCTs in other domains, such as ISCTs for pharmaceutical products.”

 I believe that the “.” before “However” should be a “;”

3. Introduction, lines 72-76: “This paper complements and builds upon related work such as [10] which provides a theoretical framing for credibility of in silico studies, [11] which discusses limitations of ASME V&V40 2018 for ISCTs, [12] which provides a framework for executing an ISCT using VVUQ concepts, and [13] which considers how to alter ASME V&V40 2018 for ISCTs, including modification of the risk assessment stage which is outside the scope of the present paper.”

 While I understand that this is a list of related work, there are too many “which”s and it becomes difficult to read.

4. Introduction, lines 76-78: “The present paper focuses primarily on first-principles models as opposed to data-driven models including machine learning based model (see following section for discussion).”

 Consider the following rewording: “The present paper focuses primarily on first-principles models as opposed to data-driven models, including those based on machine learning (see following section for discussion).”

5. Section 2, line 100: there should be a comma between “pathophysiology” and “Those” rather than a period. It sounds disjointed and awkward to read.

6. End of Section 2.2, line 232: no need to redefine “clinical trial (CT)” – similarly, check for other later redefinitions of abbreviations.

7. End of Section 2.3.2, line 278: “in silico” should be italicized

8. Beginning of Section 2.3.6, lines 331-333: The statement “The specific application considered in [26] is cardiac lead fracture, but here we focus on the statistical framework developed in [26] which is the main contribution of that paper.” is worded unclearly. It isn’t easily apparent that the statistical framework is the main contribution of [26] and it should be reworded.

9. Beginning of Section 2.3.9, line 384: I think the “e.g.,” here would read better if it was replaced with “such as” or a similar phrase. Additionally, this whole section is difficult to read because of the many references and should be made easier to follow if possible. The parentheses and such break up the sentences, which makes it so that the reader has to re-read this section multiple times to understand what is being said.

10. Beginning of Section 3.1, lines 407-408: I think that “ISCTs comprise various sub-models” should be “ISCTs are comprised of various sub-models”.

11. Next paragraph, lines 416-417: The sentence “In our previous work for patient-specific models [9] we identified unique calculation verification considerations that arise, some of these also apply to ISCTs.” is confusing to read and should be reworded.

12. Lines 427-428, when introducing the strategies: I think that “(in no specific order)” might sound better than the existing text “(the following are not ordered by rigor)”

13. Line 439: There should be a semicolon “;” instead of the existing comma “,” when introducing the example.

14. Line 471, when describing device model validation: “device/patient model” should be “device-patient model” (which I base off of the later use of “device-

patient” in the manuscript)

15. Line 541: “area under receiver operating characteristics (ROC) curve” is later abbreviated as AUC (and ΔAUC), so I would recommend sticking to AUC since ROC is not used again in the text.

16. Section 3.2.3, lines 617-619, when presenting differences in output co-variation: there should be a citation (or citations) for the women having smaller hearts and heart size vs. tissue properties statement.

17. Section 3.3., lines 644-651: I don’t think “uncertainties” for each item in this list needs to be repeated because “other sources of uncertainty” was already stated in the text when introducing the list.

18. Discussion, line 729: I think that “device/patient” should be “device-patient”

19. Discussion, line 803: “low-hanging fruit” may be a little too colloquial, I’d suggest an alternative synonym.

**Have the authors made all data and (if applicable) computational code underlying the findings in their manuscript fully available?**

Reviewer #1: Yes

Reviewer #2: **No: **Not applicable, it is a review typ

---

## [Editor Report · Decision Letter 1]

1 Jul 2024

Dear Dr. Pathmanathan,

We are pleased to inform you that your manuscript 'Credibility Assessment of In Silico Clinical Trials for Medical Devices' has been provisionally accepted for publication in PLOS Computational Biology.

Best regards,

Mukesh Dhamala

Guest Editor

PLOS Computational Biology

Marc Birtwistle

Section Editor

PLOS Computational Biology

---

## [Editor Report · Acceptance letter]

14 Jul 2024

PCOMPBIOL-D-23-01864R1 

Credibility Assessment of *In Silico* Clinical Trials for Medical Devices

Dear Dr Pathmanathan,

I am pleased to inform you that your manuscript has been formally accepted for publication in PLOS Computational Biology. Your manuscript is now with our production department and you will be notified of the publication date in due course.

With kind regards,

Zsanett Szabo
